

# Measurement report: Sources, sinks and lifetime of NO$_X$ in a sub-urban temperate forest at night

Simone T. Andersen[1], Max R. McGillen[2], Chaoyang Xue[2], Tobias Seubert[1], Patrick Dewald[1], Gunther N. T. E. Türk[1], Jan Schuladen[1], Cyrielle Denjean[3], Jean-Claude Etienne[3], Olivier Garrouste[3], Marina Jamar[4], Sergio Harb[5], Manuela Cirtog[5], Vincent Michoud[6], Mathieu Cazaunau[5], Antonin Bergé[5], Christopher Cantrell[5], Sebastien Dusanter[4], Bénédicte Picquet-Varrault[5], Alexandre Kukui[7], Abdelwahid Mellouki[2,8], Lucy J. Carpenter[9], Jos Lelieveld[1], John N. Crowley[1]

[1]Atmospheric Chemistry Department, Max-Planck-Institute for Chemistry, 55128-Mainz, Germany

[2]Institut de Combustion, Aérothermique, Réactivité Environnement (ICARE), CNRS, 1C Avenue de la Recherche Scientifique, CEDEX 2, 45071 Orléans, France

[3]CNRM, Universite de Toulouse, Meteo-France, CNRS, Toulouse, France

[4]IMT Nord Europe, Institut Mines-Télécom, Université de Lille, Center for Energy and Environment, 59000 Lille, France

[5]Univ Paris Est Creteil and Université de Paris Cité, CNRS, LISA, F-94010 Créteil, France

[6]Université Paris Cité and Univ Paris Est Creteil, CNRS, LISA, F-75013 Paris, France

[7]Laboratoire de Physique et Chimie de l'Environnement et de l'Espace (LPC2E), CNRS Orléans, France

[8]University Mohammed VI Polytechnic (UM6P), Lot 660, Hay Moulay Rachid Ben Guerir, 43150, Morocco

[9]Wolfson Atmospheric Chemistry Laboratory, Department of Chemistry, University of York, York, UK

*Correspondence to*: Simone T. Andersen (simone.andersen@mpic.de) and John N. Crowley (john.crowley@mpic.de)





## 1 Abstract

Through observations of NO, NO$_2$, NO$_Y$ and O$_3$ in the Rambouillet forest near Paris, France, (as part of the ACROSS campaign, 2022) we have gained insight into nighttime processes controlling NO$_X$ in an anthropogenically impacted forest environment. O$_3$ mixing ratios displayed a strong diel profile at the site, which was driven by a variable but generally rapid deposition to soil and foliar surfaces. The O$_3$ diel profile was strongly influenced by relative humidity, which impacted the surface resistance to uptake, and temperature inversion, which influenced the rate of entrainment of O$_3$ from above the canopy. Only when the O$_3$ mixing ratio was sufficiently low (and thus the NO lifetime sufficiently long), were sustained NO peaks observed above the instrumental detection limit, enabling derivation of average NO emission rates from the soil of ~1.4 ppbv h$^{-1}$. Observations of the lack of increase in NO$_2$ at night, despite a significant production rate from the reaction of NO with O$_3$, enabled an effective lifetime of NO$_2$ of ~0.5-3 h to be derived. As the loss of NO$_2$ was not compensated by the formation of gas- or particle-phase reactive nitrogen species it was presumably driven by deposition to soil and foliar surfaces, or any products formed were themselves short-lived with respect to deposition. By comparison, the daytime lifetime of NO$_2$ with respect to loss by reaction with OH is about 1 day. We conclude that the nighttime deposition of NO$_2$ is a major sink of boundary layer NO$_X$ in this temperate forest environment.

## 2 Introduction

Nitrogen oxides (NO$_X$ = NO + NO$_2$) are pollutant trace gases, which play a key role in the atmosphere by producing or destroying tropospheric ozone (O$_3$), which can cause respiratory illness (Ciencewicki and Jaspers, 2007) and damage to plants (Emberson et al., 2018). Photolysis of nitrogen dioxide (NO$_2$) (R1) is the primary source of tropospheric ozone (O$_3$), and the nitric oxide (NO) product is oxidized back to NO$_2$ either by O$_3$ (R2) or by organic peroxy radicals (RO$_2$, under formation of alkoxy radicals (RO)) or hydroperoxyl radicals (HO$_2$) (R3, R4) (Lightfoot et al., 1992). The latter results in formation of the hydroxyl radical (OH) radical, and R3 and R4 thus represent routes to recycle the most important atmospheric radical initiator of oxidation (Hens et al., 2014). It is, therefore, essential to understand the sources and sinks of NO$_X$ in the atmosphere.

$$NO_2 + h\nu\ (+O_2) \rightarrow NO + O_3 \qquad (R1)$$

$$NO + O_3 \rightarrow NO_2 + O_2 \qquad (R2)$$

$$NO + RO_2 \rightarrow NO_2 + RO \qquad (R3)$$

$$NO + HO_2 \rightarrow NO_2 + OH \qquad (R4)$$

The dominant global sources of NO$_X$ are anthropogenic in the form of combustion of fossil fuels and to a lesser degree biomass burning and agricultural soils. The natural sources, which include lightning (Schumann and Huntrieser, 2007), wildfires (Val Martin et al., 2008), and unperturbed soil emissions from microbial activities (Davidson and Kingerlee, 1997), are important in regions remote from anthropogenic sources. NO$_2$ and NO both react with peroxy radicals in the atmosphere



to produce organic nitrates (R5-R6), including peroxy nitrates ($RO_2NO_2$) and alkyl nitrates
($RONO_2$), which are important precursors for the formation of secondary organic aerosols (SOA)
(Hallquist et al., 2009; Kanakidou et al., 2005; Kiendler-Scharr et al., 2016). $NO_2$ also reacts with
OH radicals, $O_3$ and nitrate radicals ($NO_3$) to form nitric acid ($HNO_3$) (R7), $NO_3$ radicals (R8), and
dinitrogen pentoxide ($N_2O_5$) (R9), respectively. $N_2O_5$ is in thermal equilibrium with $NO_2$ and $NO_3$
and can interact with aqueous aerosol or moist surfaces to form $HNO_3$ (R10) (Kane et al., 2001)
or nitryl chloride ($ClNO_2$) (Phillips et al., 2013; Phillips et al., 2012). Organic nitrates, SOA, and
$HNO_3$ are all removed from the boundary layer through dry and wet deposition which thereby
removes $NO_X$ from the atmosphere.
$NO_2 + RO_2 + M \rightarrow RO_2NO_2 + M$                                                        (R5)
$NO + RO_2 + M \rightarrow RONO_2 + M$                                                          (R6)
$NO_2 + OH + M \rightarrow HNO_3 + M$                                                            (R7)
$NO_2 + O_3 \rightarrow NO_3 + O_2$                                                                      (R8)
$NO_2 + NO_3 + M \rightleftharpoons N_2O_5 + M$                                                      (R9)
$N_2O_5 + H_2O\ (aq) \rightarrow 2\ HNO_3\ (aq)$                                                     (R10)
In the planetary boundary layer, $NO_2$ is also lost through dry deposition to surfaces such as soil
and leaves. Deposition takes places both at nighttime and daytime, but is expected to be more
efficient during daytime due to increased mixing through turbulence. When $NO_2$ deposits onto
humid surfaces, it can lead to the production of nitrous acid (HONO), which can be released to the
atmosphere (Meusel et al., 2016; Elshorbany et al., 2012). $NO_2$ uptake on leaves takes place
through stomatal and non-stomatal processes, which have been reported to depend on multiple
factors such as stomata aperture and relative humidity. Stomatal uptake primarily occurs at
daytime when the stomata are open, leading to increased $NO_2$ loss compared to nighttime, when
the stomata are not fully open (Delaria et al., 2020; Delaria et al., 2018; Chaparro-Suarez et al.,
2011). Non-stomatal uptake occurs through the cuticles, though the importance of cuticular uptake
has been reported to be small compared to the stomatal uptake (Delaria and Cohen, 2020; Delaria
et al., 2020). $NO_2$ uptake to leaves is reported to be enhanced in the presence of water films, which
may exist when the relative humidity is >70% (Thoene et al., 1996; Weber and Rennenberg, 1996;
Burkhardt and Eiden, 1994). There is, however, no consensus on this process, as other studies have
not observed this effect (Gessler et al., 2000). Most recent work shows that the interactions with
foliar surfaces is uni-directional, i.e. emissions are negligible (Delaria et al., 2020).
At nighttime, $NO_2$ photolysis ceases and as a consequence, in the absence of combustion sources,
the main sources of NO are emissions from soils (Jaeglé et al., 2005). Since NO is oxidised
efficiently by $O_3$ at night, its concentration will be highest at the surface and will decrease with
altitude. The vertical profile of $O_3$ is the opposite owing to its physical loss due to deposition near
the surface and through chemical reaction with NO and/or alkenes combined with entrainment
from the nocturnal residual layer. As $NO_2$ is produced from the reaction between NO and $O_3$, its



vertical gradient is expected to be weaker than those of NO and $O_3$ (Geyer and Stutz, 2004; Stutz
et al., 2004).
In this study we use measurements from the ACROSS (Atmospheric ChemistRy Of the Suburban
foreSt) campaign to investigate the nighttime sources and sinks of $NO_X$ in a temperate forest. $O_3$
measurements are used to explain the observed NO features and measurements of $NO_2$ and total
gas-phase nitrogen species ($NO_Y$) and particulate nitrate are used to investigate the lifetime and
fate of $NO_X$ in the forest environment.

**3   The ACROSS Campaign**
The ACROSS campaign (13th of June 2022 to 25th of July 2022) was conducted in multiple
locations in and around Paris, France (Cantrell and Michoud, 2022). Here we present
measurements from the Rambouillet forest supersite located approximately 50 km southwest of
Paris (48.687, 1.704). The forest consists of approximately 70% oak, 20% pine, and small
contributions from beech and chestnut. The top of the forest canopy around the supersite was
around 20-25 m. Several instrumented containers were placed in a clearing (~697 $m^2$) together
with a 41 m measurement tower. Most of the instruments used in this study were located in two
different containers (MPIC and Orléans). The sampling inlets of the two containers were
approximately 17 m apart and the tower was approximately 9 m from the MPIC container and 16
m from the Orléans container. The soil measurements were carried out at the bottom of the tower,
approximately 13 m from the MPIC container and approximately 17 m from the Orleans container.
All the instruments used in this study are described briefly below.

**3.1   Measurements**
**3.1.1   Ground**
$NO_2$ was measured using two different cavity ringdown spectroscopy (CRDS) instruments with
co-located inlets sampling from a high-volume-flow stainless steel tube (10 $m^3\,min^{-1}$; 15 cm
diameter, 0.2 s residence time) taking air from a height of 5.4 m above ground. One of the
instruments (5CH-CRDS) consists of 3 cavities operated at 408 nm to measure $NO_2$ and, via their
thermal dissociation to $NO_2$, total peroxy nitrates ($\sum$PNs, 448 K) and total alkyl nitrates ($\sum$ANs,
648 K). Two additional cavities, operated at 662 nm, measured $NO_3$ and (via thermal dissociation
to $NO_3$, 373 K) $N_2O_5$ (Sobanski et al., 2016). During this campaign, the $NO_2$ cavity had a limit of
detection (LOD) of 9.7 pptv for 1 min averaging (3$\sigma$). The second instrument (k-NO3) primarily
measures the $NO_3$ reactivity, but also has a cavity operated at 405 nm for the measurement of $NO_2$
(Liebmann et al., 2018).
Another CRDS instrument was used to measure $NO_X$, $NO_Y$, and particulate nitrate ($pNO_3$) from
co-located inlets near the high-volume-flow stainless steel tube. $NO_X$ was measured by adding $O_3$
to the ambient sample, thereby oxidizing NO to $NO_2$, which was measured with CRDS at 405 nm
(Friedrich et al., 2020). A judicious choice of $O_3$ and reaction time ensured that minimal (>1%) of
$NO_2$ was oxidized to $NO_3$. At times with low (or zero) NO, $NO_X$ concentrations were in close



agreement with both $NO_2$ measurements. $NO_Y$ was measured by passing ambient air through a
quartz inlet at ~ 900 K which quantitatively converts reactive nitrogen trace-gases to NO or $NO_2$.
Exceptions are $N_2O$, HCN and $NH_3$, which are not detected. In this location, $NO_Y$ is expected to
consist mainly of $NO_X$ + $NO_3$ + $N_2O_5$ + $HNO_3$ + PNs + ANs + HONO + $ClNO_2$ + particulate
nitrates ($pNO_3$).
Particulate nitrates (both organic and inorganic) were separately measured (as $NO_Y$) after denuding
gas-phase reactive nitrogen species (Friedrich et al., 2020). To achieve this, problems involving
the ineffective trapping of gas-phase $NO_X$ by the denuder was eliminated, as will be described in
a forthcoming technical paper.
$O_3$ was measured from the high-volume-flow stainless-steel tube with a commercial instrument
(2B Technologies model 205) using UV absorption at 254 nm. The LOD is 2 ppbv for 10 s
averaging time.
A spectral radiometer (metcon Gmbh) was installed near the co-located inlets on top of the MPIC
container to measure actinic fluxes, which were used to calculate photolysis frequencies as
described elsewhere (Meusel et al., 2016).
NO was measured from the Orléans container using a commercial chemiluminescence instrument
(Ecophysics CLD 780 TR, henceforth CLD) with an LOD of 10 pptv for 1 min averaging time.
The sampling height for NO measurements was about 0.6 and 3.2 m above the container top and
the ground surface, respectively. The NO measurements required correction due to a change in the
CLD sensitivity during the campaign caused by an interruption in the instrument's oxygen supply.
The corrections and the corrective procedure are described in the SI.
HONO was measured by a commercial long-path absorption photometer (LOPAP-03, QUMA
GmbH, Germany) with a sampling height of 2.0 m above the ground level. Details about the
LOPAP instrument can be found elsewhere (Heland et al., 2001; Kleffmann et al., 2006). During
the campaign, the LOPAP was calibrated by diluted nitrite when changing any supporting
solutions. Zero calibration by measuring synthetic air was conducted 2-3 times per day. The
detection limit is < 5 pptv.
The sum of peroxy radicals, $XO_2$=$HO_2$+$RO_2$, was measured by their conversion to $H_2SO_4$ in
presence of NO and $SO_2$ and detection of the generated $H_2SO_4$ using $NO_3^-$ CIMS (Kukui et al.,
2008). The calibration coefficient is determined using $N_2O$ actinometry and OH/$RO_2$ generation
in a turbulent flow reactor by photolysis of $N_2O$ or $H_2O$ at 184.9 nm. The calibration of $HO_2$,
$CH_3O_2$ and other $RO_2$ is performed by adding into the calibration reactor CO, $CH_4$ (or other $RO_2$
precursors) converting OH to $RO_2$. The overall estimated calibration accuracy ($2\sigma$) for $XO_2$ is
about 30%, although the uncertainty of the $XO_2$ measurements is typically higher due to
uncertainty in ambient air $XO_2$ composition. The lower limit of detection for $XO_2$ radicals at S/N=3
and a 4 minute integration time is $2\times10^6$ molecule $cm^{-3}$.
Time series of the most relevant measurements can be found in Figure S1-2. Due to missing total
$NO_X$ and $NO_Y$ measurements prior to June 25[th] and NO after July 18[th], the data analysis is focused
on the time period in between these dates.



### 3.1.2   Tower

Measurements at 41 m were conducted with instruments located on the tower as well as through a manifold with an inlet at the top of the tower. The manifold was built from glass tubing (4.9 cm inner diameter, Borodrain) with a residence time in the manifold of 2.1 s. $NO_2$ was measured using a cavity attenuated phase shift (CAPS) instrument on the tower with an LOD of 40 pptv, which was zeroed every 1-2 hours. NO and $O_3$ were both measured from the manifold using a chemiluminescence instrument with a LOD of 30 pptv and a HORIBA (APOA370) with an LOD of 2.5 ppbv, respectively. The NO measurements were corrected for losses due to the reaction of NO with $O_3$ in the manifold and the sampling line (total 5.5 s), with corrections ranging from 1-28%. Time series of all three measurements are plotted in Figure S3.

### 3.1.3   Meteorology and Soil

Ambient temperature was measured at four different heights on the tower; 5 m, 13 m, 21 m, and 41 m using temperature sensors from Atexis (PT1000) and Thermoest (PT100). Relative humidity was measured at 5 m using a Vaisala humidity sensor (HMP45A). Soil temperature and moisture were measured at 5 cm, 10 cm, and 30 cm below the surface using probes from Thermoest (PT100) and Delta T (Thetaprobe ML2X), respectively. Wind speed and direction were measured at 41 m using a wind monitor from Young Company. Time series of all the meteorological and soil measurements are shown in Figure S4-5.

### 3.2   HYSPLIT

To identify different air masses, 48-hour back trajectories were simulated every hour at a terminating height of 40 m using the Hybrid Single-Particle Lagrangian Integrated Trajectory model (HYSPLIT, version 4, 2019) (Draxler and Rolph, 2011). The back-trajectories were modelled using meteorological data from the Global Data Assimilation System (GDAS) at a resolution of 1 degree. This led to the separation of the data into two periods, 25[th] of June to 2[nd] of July and 3[rd] of July to 18[th] of July, which are plotted in Figure 1. The first phase is dominated by clean air from over the Atlantic Ocean (henceforth called "Atlantic"). Back trajectories indicated that the vast majority of air masses were transported within the boundary layer prior to reaching the site and thus may have reasonably fresh "marine influence". The second phase is dominated by air that has passed over urban locations including Paris, Brussels and the Ruhr area within the last 48 hours (henceforth called "Continental").

### 4   Results and Discussion

Two 24-hour periods of temperature (at 4 different heights), NO, $O_3$, relative humidity (RH), $NO_2$, and $NO_2$ photolysis rate constant ($JNO_2$) are plotted in Figure 2. The left panels show 24 hours with Atlantic air and the right panels 24 hours with continental air. Immediately apparent in these datasets (and in Fig S1) is the large diel cycle in $O_3$ mixing ratios, with net daytime production



resulting in mid-afternoon mixing ratios between ~30 and 90 ppbv. In contrast, very low $O_3$ mixing
ratios (often approaching zero) were observed at nighttime.
In the lowermost panels ($JNO_2$ measurements), the nighttime is shown in dark grey and the two
light grey areas show the time before sunset (about 5 hours) and after sunrise (about 4.5 hours)
when very little direct sunlight reaches the ground of the site due to shading by the trees. This
leaves about 6.5 hours centred around midday when direct sunlight reaches the ground. The
shading results in radiative cooling of the ground in the late afternoon and associated temperature
inversions begin to form prior to sunset as can be observed in the right panels of Figure 2 and in
more detail in Figure S6. The temperature inversions begin approximately at the same time as the
ground temperature at 5 cm below the surface starts to decrease (see Figure S6). These conditions
of insolation were relatively consistent throughout the campaign.
Clear temperature inversions were observed for both nights shown in Figure 2, the beginning and
end of which are indicated by dashed lines. Vertical mixing can be reduced significantly during a
temperature inversion, which is apparent from the $O_3$ and RH measurements in the right-hand
panel. In both examples, $O_3$ decreases at the ground level (5.4 m) at the beginning of the
temperature inversion and increases as the inversion breaks down in the morning. This behaviour
is understood in terms of $O_3$ loss to soil surfaces and through stomatal and non-stomatal uptake on
leaves (Zhou et al., 2017; Rannik et al., 2012; Altimir et al., 2006; Ganzeveld and Lelieveld, 1995)
as well as through chemical reactions with e.g. NO, $NO_2$ and unsaturated (biogenic) organics
(Kurpius and Goldstein, 2003). Reduced vertical mixing means that during the inversion, $O_3$ is
only slowly replenished by downward mixing of air masses above the canopy where higher $O_3$
levels are observed. In contrast, the RH behaves in the opposite sense as the air above the inversion
is drier than close to the ground, where evapotranspiration contributes to enhanced water vapour
concentrations.
If the only source of NO was the photolysis of $NO_2$, NO mixing ratios would be expected to follow
the $NO_2$ photolysis rate during the day and tend to zero at night as NO is oxidized on a time scale
of minutes (for $O_3 > 10$ ppb) to $NO_2$ by $O_3$. This was not always the case during ACROSS. A
pronounced NO peak (up to ~2 ppbv) was observed at ground level between 00:00 and 06:00 UTC
(02:00 and 08:00 local time) during the phase dominated by Atlantic air, shown in Figure 2, which
is absent in the phase dominated by continental air. The peak occurs prior to sunrise and is only
observed by the ground-level measurements suggesting a non-photolytic source of NO close to the
ground, which is discussed further below. Very low (0-5 ppbv) $O_3$ mixing ratios coincide with the
sustained nighttime NO peak observed, which is never reached in the example from the continental
phase, although in both cases clear temperature inversions were seen. Additional examples of
sustained NO peaks (i.e. lasting several hours at level between 1 and 2 ppbv) at night during the
first phase are shown in Figure S7. Examples of additional nights with temperature inversions
during phase 2, where NO mixing ratios remained close to zero, are shown in Figure S8.



### 4.1  Nighttime Ozone Loss

For each night between June 17[th] and July 22[nd] the net $O_3$ loss rate constant, $k_L(O_3)$ was derived
by fitting exponential expressions to the data for periods of 4.5 to 8 hours. $k_L(O_3)$ was highly
variable, with values between $1.8 \times 10^{-5}$ s$^{-1}$ and $3.0 \times 10^{-4}$ s$^{-1}$, depending on the strength of the
temperature inversion and the relative humidity (see discussion below). These values of $k_L(O_3)$
correspond to lifetimes of 1-15 hours for $O_3$ at nighttime. Chemical losses of $O_3$ occur through
reactions with NO, $NO_2$, and unsaturated BVOCs (Zhou et al., 2017). Rate coefficients of reactions
of $O_3$ with NO ($1.9 \times 10^{-14}$ cm$^3$ molecule$^{-1}$ s$^{-1}$ at 298 K), $NO_2$ ($3.5 \times 10^{-17}$ cm$^3$ molecule$^{-1}$ s$^{-1}$ at 298
K), limonene (a reactive terpene, $2.2 \times 10^{-16}$ cm$^3$ molecule$^{-1}$ s$^{-1}$ at 298 K), β-caryophyllene
(sesquiterpene, $1.2 \times 10^{-14}$ cm$^3$ molecule$^{-1}$ s$^{-1}$ at 298 K) are low such that mixing ratios in excess
of 1 ppbv for NO and β-caryophyllene would be required to explain the $O_3$ loss rate constant
(IUPAC, 2023). Required mixing ratios of terpenes or $NO_2$ would be even larger (60-300 ppbv).
As such high mixing ratios of NO and $NO_2$ were not observed continuously and such levels of
BVOC are unlikely, we assume that chemical losses of $O_3$ are insignificant compared to deposition
as previously observed (Zhou et al., 2017). Ignoring entrainment from other heights, we can then
equate $k_L(O_3)$ to ($2V_d/h$), where $V_d$ is the deposition velocity and $h$ is the boundary layer height;
the factor 2 is used to account for a positive vertical gradient (Shepson et al., 1992). Using a
boundary layer height of 20 m (arbitrarily set equal to the top of the canopy) gives net deposition
velocities varying between 0.018 and 0.3 cm s$^{-1}$. These values for $V_d$ are in broad agreement with
other studies in temperate forests, where deposition velocities for $O_3$ at nighttime have been
reported to be around 0.07-0.3 cm s$^{-1}$ (Padro, 1996, 1993; Finkelstein et al., 2000; Wu et al., 2016).
In Figure 3 the $O_3$ production rate ($JNO_2 \times [NO_2]$), RH, temperature at 4 different heights and $O_3$
mixing ratio have been plotted for two nights with high average RH to illustrate the impact of
temperature inversions on the net $O_3$ loss-rate constants. The production rate of $O_3$ is used to
identify periods in which production is negligible. In the left panel a night without a temperature
inversion is plotted, where the average RH for the period used to fit the exponential decay is 93 ±
3 %. These conditions resulted in a net $O_3$ loss-rate constant of $6.0 \times 10^{-5}$ s$^{-1}$. In contrast, the night
depicted in the right panel has the same average RH (92 ± 3 %) and a very clear temperature
inversion, which gives a net $O_3$ loss-rate constant of $3.0 \times 10^{-4}$ s$^{-1}$. This gives a factor of 5 between
these two net $O_3$ loss-rate constants depending on whether a temperature inversion is observed or
not. This can be understood in terms of the $O_3$ being replenished from above when there is no (or
a weak) inversion, which is not the case when there is an inversion. Bearing this in mind, the use
of $k_L(O_3)$ (a *net* $O_3$ loss constant) must result in a lower limit to $V_d$ unless strong temperature
inversions (preventing $O_3$ entrainment from above) are present. The $O_3$ loss rate will also be
enhanced under conditions of strong inversion if trace-gases that are reacive towards $O_3$ are
released into a very shallow boundary layer. However, as indicated above, chemical losses are not
expected to compete with physical losses.
To investigate the impact of RH on the net $O_3$ loss-rate constants, two nights with temperature
inversions are plotted in Figure S9; one with high RH (92 ± 3 %) and one with a lower RH (63 ±
6 %). Here we see a large decrease in $k_L(O_3)$ from $3.0 \times 10^{-4}$ s$^{-1}$ to $4.5 \times 10^{-5}$ s$^{-1}$, when going from
high to lower RH. The individually determined $O_3$ loss-rate constants are plotted as a function of
RH in Figure 4 and coloured depending on whether a temperature inversion is observed or not





during the time period which was used for the exponential decay fit. A clear increase in $O_3$ loss-
rate constants can be observed when RH increases above 70-80% when a temperature inversion
was observed. A small increase at RH higher than 70-80% was also observed when temperature
inversions were absent. The observed dependence of $k_L(O_3)$ on relative humidity is consistent with
previous studies in forested regions, which have reported an increase in $O_3$ loss above 60-70% RH
(Altimir et al., 2006; Rannik et al., 2012; Zhou et al., 2017). Altimir et al. (2006) suggested an
enhancement factor which is humidity dependent above 70% RH; 1 at 70% RH, 2 at 85% RH and
a sharp increase to over 5 when moving towards 100% RH. In a boreal forest these observations
have been explained by the formation of a "wet skin" on leaves which enhances surface $O_3$ losses
by modifying (reducing) the surface-resistance to uptake (Zhou et al., 2017). This is in broad
agreement with our observations during nights with a temperature inversion (see Figure 4), and
the discrepancies between the studies could be explained e.g. by different tree types, the height of
the boundary layer, strength of the inversion and temperature.
The faster net rate of $O_3$ loss on nights with high relative humidity and well-defined temperature
inversions explain the differences observed in the $O_3$ mixing ratios at night during the Atlantic and
continental phases. The average nighttime (20:00-04:00 UTC) RH for the Atlantic phase was 87.4
± 7.6 (1σ) % compared to 68.4 ± 12.7 (1σ) % for the continental phase, indicating that on nights
with temperature inversions higher loss-rate constants would be expected for the Atlantic phase.
The high RH combined with the significantly lower average peak $O_3$ mixing ratio in the Atlantic
phase (34.5 ± 6.0 (1σ) ppbv between 14:00-15:00 UTC) compared to the continental phase (52.7
± 13.6 (1σ) ppbv between 14:00-15:00 UTC) explains why on nights with temperature inversions
during the Atlantic phase the $O_3$ was essentially completely depleted as shown in Figure 2 and S7.

### 4.2 Nitrogen Oxide Soil Emissions

Figure 2 (left panel) and S7 show nighttime periods in which NO was observed when $O_3$ was
depleted during the Atlantic phase. The several hours duration of the period when NO was above
the LOD excludes very local combustion as the source, leaving soil emissions resulting from
microbial activity (Davidson and Kingerlee, 1997) as the most likely source of NO. At 293 K and
2 ppbv of $O_3$, the lifetime of NO towards reaction with $O_3$ is around 20 minutes. It is therefore
reasonable to assume that NO is close to steady-state when there is 2 ppbv or more of $O_3$ available.
The NO emission rate ($E_{NO}$) can therefore be equated to the loss rate of NO as described in equation
(1) assuming all peroxy radicals ($XO_2$) react with the same rate coefficient as $HO_2$:
$$E_{NO} = k_{NO+O3}[NO][O_3] + k_{NO+HO2}[XO_2][NO] \tag{1}$$
where $k_{NO+O3}$ and $k_{NO+HO2}$ are the temperature-dependent rate constants for the reaction between
NO and $O_3$ and $HO_2$, respectively, (IUPAC, 2023) and [NO], [$O_3$] and [$XO_2$] are the measured
concentrations of NO, $O_3$ and $XO_2$, respectively. In Figure 5, NO and $E_{NO}$ (when $O_3 > 2$ ppbv) at
nighttime ($JNO_2 < 10^{-5}$ s$^{-1}$) are separated by air masses and plotted against $O_3$, where the outliers
are defined as being outside $1.5 \times$ interquartile range (IQR). While the nighttime NO mixing ratio
increased rapidly when $O_3$ tended towards 0 ppbv during the Atlantic phase, $O_3$ was never depleted
to less than 5 ppbv during the Continental phase and therefore no sustained periods of enhanced



NO were observed at nighttime. In contrast, no significant trend is found when plotting $E_{NO}$ against
$O_3$ for either of the phases, which shows that the calculated soil emission of NO is not dependent
on $O_3$. This indicates that while the soil is an important but variable source of NO, sustained
nighttime NO peaks are only observed above the instrument LOD when $O_3$ is almost totally
depleted so that the lifetime of NO is long enough to allow its concentration to build-up
sufficiently.
Water content and temperature have previously been shown to impact the emission rate of NO
from soil (Pilegaard, 2013; Rosenkranz et al., 2006). Rosenkranz et al. (2006) found a positive
correlation between soil moisture and NO emission up to 40% water-filled pore space (WFPS) and
an optimum between 12.5 and 15 °C soil temperature in a sessile oak forest in Hungary. In Figure
6, NO and $E_{NO}$ are plotted against the soil temperature and moisture at 5 cm below the surface.
The measured NO mixing ratios peak towards the highest soil moisture and lowest soil temperature
measured during this campaign, however, as with $O_3$, there is no significant trend in the NO
emission rates with soil moisture. At the low (11.5-12.5 °C) and high (19.5-20.5 °C) nighttime soil
temperatures very few measurements were made (around 2 hours combined) compared to the rest
of the temperature intervals. Across the remaining temperature intervals, no significant trend was
observed in the estimated NO emission.
The average NO emission rate derived for the two phases is identical with values of
$1.45 \pm 1.61$ ppbv h⁻¹ (1σ, median = 1.27 ppbv h⁻¹) and $1.42 \pm 5.68$ ppbv h⁻¹ (1σ, median = 0.71
ppbv h⁻¹) for the Atlantic and Continental phases, respectively, when using data where $O_3 > 2$
ppbv. The Continental phase show much higher variability resulting from more spikes in the data
during that period. When $O_3$ is completely depleted during the Atlantic phase, the increase in NO
per hour results in NO emission rates of 0.3-1.8 ppbv h⁻¹, which is in reasonable agreement with
the averages across each of the two phases when there is still $O_3$ present. By assuming a mixed
nocturnal boundary layer (NBL) with a height of 20 m (top of the canopy), the average emission
rates can be converted to NO emission fluxes of $16.6 \pm 18.5$ (1σ) and $16.2 \pm 65.0$ (1σ) µg N m⁻² h⁻
¹, respectively. These values are within the range of previous measurements in different European
forests with similar tree types to those found in the Rambouillet forest (see Table 1). The
measurements by Pilegaard et al. (2006) and Rosenkranz et al. (2006) were all performed using
the chamber technique, whereas Schindlbacher et al. (2004) measured the emission from soil
samples collected in the field and exposed to different temperatures and humidity in the laboratory.
The chamber-derived emission rates are all either lower or, within combined uncertainties, equal
to the values determined in this study, while emission rates from the soil samples were higher than
(or, within combined uncertainties, equal to) the values derived in the present study. Davidson and
Kingerlee (1997) modelled the global NO emission inventory from soil depending on the biome
(e.g. temperate forest, agriculture, and savanna), and split the temperate forest category into
regions affected by nitrogen deposition or not. For temperate forests not affected by nitrogen
deposition, those authors estimated a flux of 0.0-0.2 kg N ha⁻¹ yr⁻¹ (0.0-2.3 µg N m⁻² h⁻¹), which is
in good agreement with the lower measurements by Pilegaard et al. (2006). In contrast, the
temperate forests impacted by nitrogen deposition had estimated fluxes of 1.1-5.0 kg N ha⁻¹ yr⁻¹
(12.6-57.1 µg N m⁻² h⁻¹), which is in good agreement with our measurements at Rambouillet where
nitrogen deposition is enhanced by pollution arriving from Paris and other surrounding urbanized



/ industrialized areas. While noting that our fluxes are broadly consistent with previous
measurements, we recognise that the calculations are based on the assumptions of a well-mixed
boundary layer of fixed height arbitrarily set at 20 m and should not be over-interpreted.

### 4.3 Nitrogen Dioxide Losses

At nighttime, in the absence of its photolysis, $NO_2$ may be expected to increase in concentration
(via R2) when a constant NO source exists (e.g. from soil, as observed here) and when $O_3$ is
present. For both the Atlantic and the Continental phases an average diel profile between 20.00
and 04.00 UTC of $NO_2$ (black) is plotted in Figure 7. No obvious increase in $NO_2$ can be observed
in the Atlantic phase and an average increase of around 1 ppbv can be observed in the Continental
phase. The expected $NO_2$ resulting from the $NO + O_3$ reaction if there were no loss mechanisms
of $NO_2$ is plotted in red. This is determined using the $NO_2$ measured at 20.00 UTC and
incrementing this value by the $NO_2$ that would have been produced through NO oxidation by $O_3$
and peroxy radicals in each time step. In both phases, the simple assumption of nighttime $NO_2$
production through $NO + O_3$ and $NO + XO_2$ and no $NO_2$ loss results in significant generation of
$NO_2$ with an overestimation of 10-12 ppbv of $NO_2$ at the end of the night compared to the measured
$NO_2$. A loss mechanism of around 1.4 ppbv $h^{-1}$ of $NO_2$ is therefore necessary to explain the
observed (lack of increase in) $NO_2$.

### 4.3.1 Chemical Losses

While during the daytime $NO_2$ is removed in a largely irreversible process through reaction with
OH radicals to form $HNO_3$, this is unlikely to represent a significant sink at nighttime. In the
absence of photochemical formation pathways, OH is generated at night in the ozonolysis of
olefins and in the reaction of $HO_2$ with $NO_3$ and NO.
$O_3 + {>}{=}{<} \rightarrow \rightarrow OH$                                                 (R11)
$NO_3 + HO_2 \rightarrow OH + NO_2 + O_2$                          (R12)
$NO + HO_2 \rightarrow OH + NO_2$                                    (R4)
In the forested environment in summer, the emissions of biogenic volatile organic compounds
(BVOC) (e.g. olefinic terpenoids) will favour R11 and simultaneously disfavour R12 as $NO_3$ will
be reduced in concentration through its reactions with BVOCs. During the ACROSS campaign
ground $NO_3$ levels were generally below instrument detection limits of 2 pptv and we can
reasonably ignore R12. Measurements of OH in forested environments are sparse, though they
indicate that nocturnal OH levels are low, with concentrations generally lower than $1 \times 10^5$
molecule $cm^{-3}$. Combining the rate coefficient for reaction of OH with $NO_2$ of $\sim 1 \times 10^{-11}$ $cm^3$
$molecule^{-1}$ $s^{-1}$ (IUPAC, 2023) at ambient pressure and $\approx 300$ K with an upper limit (confirmed by
measurements) to the OH concentration of $1 \times 10^6$ molecule $cm^{-3}$ results in a $NO_2$ loss constant of
$1 \times 10^{-5}$ $s^{-1}$, or (at the average nighttime $NO_2 = 1650$ pptv) a loss rate of $\sim 60$ ppt $h^{-1}$, clearly
insufficient to explain the observations.



$NO_2$ is also lost via its reaction with $O_3$ to form the $NO_3$ radical (R8). In an upcoming paper, we
will show that the majority of $NO_3$ formed in the forest will react with BVOCs rather than with
NO (to re-form $NO_2$) and, to a good approximation, R8 represents an irreversible loss of $NO_2$ as
the alkyl nitrates will not release nitrogen in the form of $NO_2$ at nighttime. However, the rate
coefficient for this process ($3.5 \times 10^{-17}$ $cm^3$ molecule$^{-1}$ s$^{-1}$ at 298 K, (IUPAC, 2023)) is very small
and with average nighttime $O_3$ levels reduced by deposition (see above) to 23 ppbv, the lifetime
of $NO_2$ with respect to this reaction is 14 hours and the loss-rate (at the average nighttime $NO_2$ =
1650 pptv) is ~120 pptv h$^{-1}$, again too slow to contribute significantly to the apparent loss rate of
$NO_2$.
The chemical loss of $NO_2$ via reaction with OH or via formation of $NO_3$ and its further reactions
with BVOC to form alkyl nitrates is expected to result in the conversion of $NO_X$ to $NO_Y$. As
described in section 3.1.1, during the ACROSS campaign we operated a $NO_Y$ instrument to
measure $NO_Y$ both in the gas- and particle-phases. Figure 8 displays the average diel profiles of
$NO_Z$ ($NO_Y$-$NO_X$) and $pNO_3$ during the Atlantic and Continental phases. For both $NO_Z$ and $pNO_3$
the diel profiles show either a decrease or stable mixing ratio across the period in which losses of
10-12 ppbv of $NO_2$ are required to explain the observations. Clearly, the loss of $NO_2$ at nighttime
is not balanced by the formation of other forms of reactive nitrogen that were long lived enough
to be detected. Trace gases such as $HNO_3$ or alkyl nitrates may be lost via deposition to surfaces,
especially at high relative humidity and lifetimes for biogenic alkyl nitrates of a few hours have
been reported (Liebmann et al., 2019; Farmer and Cohen, 2008; Browne et al., 2013; Romer
Present et al., 2019). However, as shown above, the limiting step in the formation of organic
nitrates is the slow reaction of $NO_2$ with $O_3$, which will not convert sufficient $NO_2$ to $NO_Z$ to
explain our observations. Formation of organic nitrates that do not require the intermediacy of $NO_3$
(i.e. peroxy nitrates formed from $RO_2$ + $NO_2$) would also have been detected by the $NO_Y$
instrument and can thus also be ruled out as major reservoirs of $NO_X$.
$NO_2$ deposited to humid surfaces can be converted to HONO and released to the atmosphere
(Elshorbany et al., 2012; Meusel et al., 2016). A time series of HONO can be found in Figure S2
which reveals increases in HONO at nighttime. However, the HONO mixing ratios can account
for only a small fraction of the $NO_2$ loss described above. This may reflect the fact that, if formed
at a moist surface, (soluble) HONO is unlikely to desorb quantitatively into the gas-phase. The
low HONO mixing ratios measured during the Atlantic phase compared to the Continental phase,
could potentially be explained by the difference in soil humidity, however, the factors influencing
the formation and release of HONO are complex. The HONO observations will be analysed in
detail in a seperate publication from the ACROSS campaign.
In the absence of other known gas-phase mechanisms for the removal of $NO_2$ at night and the fact
that very little other reactive nitrogen trace-gases or nitrate particles are formed during the night,
we conclude that physical removal of $NO_2$ (i.e. deposition) is responsible for its lack of build-up
at night during ACROSS and that any transformation of $NO_2$ at the surface does not lead to
quantitative release into the gas-phase





### 4.3.2 Physical Losses


$NO_2$ is known to be lost through dry deposition to surfaces such as soil and leaves, the latter
depending on whether the stomata are open (daytime) or not fully open (nighttime) (Delaria et al.,
2020; Delaria et al., 2018; Chaparro-Suarez et al., 2011). As for $O_3$, dry deposition of $NO_2$ to
surfaces can be described by an exponential decay with a first-order decay rate constant, $k_L(NO_2)$
$= (V_d/h)$, where $V_d$ is the deposition velocity and $h$ is the boundary layer height. This expression
applies when gradients within the boundary layer are weak, as expected for $NO_2$ (see above) even
though vertical mixing is very slow at night. The net production (or loss) of $NO_2$ is given by Eq.
(2) where the first term on the right-hand side is the $NO_2$ production rate from the reaction of NO
with $O_3$ or $XO_2$ (which is identical to the NO soil emission rate) and the second term is the loss
rate assuming only depositional losses (see above) and ignoring entrainment of $NO_2$ from other
heights. This will give an upper limit of the $NO_2$ deposition rate as a small fraction (<10%) of $NO_2$
is lost through chemical reactions with $O_3$ and OH (see above).
$$\frac{d[NO_2]}{dt} = E_{NO} - k_L(NO_2)[NO_2]_0 \tag{2}$$

$[NO_2]_0$ is the $NO_2$ mixing ratio at 20.00 UTC. The $NO_2$ concentration at any subsequent time can
then be calculated as described in Eq. (3) with variation of $k_L(NO_2)$ in order to match the observed
$NO_2$ mixing ratio.
$$[NO_2]_t = \int_0^t \frac{d[NO_2]}{dt} + [NO_2]_0 \tag{3}$$

In Figure 7 the grey lines symbolize the calculated $NO_2$ mixing ratios at nighttime using values of
$k_L(NO_2)$ between $1.0 \times 10^{-4}$ and $4.0 \times 10^{-4}$ s$^{-1}$. As expected, no single value of $k_L(NO_2)$ can explain
all the measurements as the height of the BL will not be invariant during the whole night. However,
for the Continental and Atlantic phases the observed $NO_2$ can be explained with $k_L(NO_2) = (2.0 \pm
1.0) \times 10^{-4}$ s$^{-1}$ and $k_L(NO_2) = (2.75 \pm 1.25) \times 10^{-4}$ s$^{-1}$, respectively, which results in lifetimes of ~
1-3 h and ~ 40-110 min for $NO_2$ at nighttime. As deposition of $NO_2$ in this environment represents
a permanent loss of $NO_X$ from the gas phase, this lifetime can be compared to e.g. the lifetime of
$NO_X$ with respect to its conversion to $HNO_3$ via reaction of $NO_2$ with OH which is ~ 1 day
(assuming average [OH] = $1 \times 10^6$ molecule cm$^{-3}$). Clearly, nighttime depositional losses of $NO_2$
in a forested environment contribute substantially to its lifetime and to the $NO_X$ budget.
If we continue to assume the nocturnal boundary layer at the forest site is at the top of the canopy
(20 m), then the $NO_2$ loss-rate constants we determined can be converted to a deposition velocity
of $0.4 \pm 0.2$ cm s$^{-1}$ and $0.55 \pm 0.25$ cm s$^{-1}$ for the Continental and Atlantic phase, respectively.
These are comparable to previous measurements of $NO_2$ deposition velocities of 0.15 cm s$^{-1}$
(Dewald et al., 2022), 0.1-0.57 cm s$^{-1}$ (Rondón et al., 1993), 0.098 cm s$^{-1}$ (Breuninger et al., 2013),
0.2-0.5 cm s$^{-1}$ (Horii et al., 2004), 0.02-0.64 cm s$^{-1}$ (Puxbaum and Gregori, 1998), for a mountain
observatory surrounded by coniferous trees, boreal coniferous forests, a temperate coniferous
forest, a temperate mixed deciduous forest, and a temperate oak forest, respectively, where a
combination of soil and foliage deposition is measured. Horii et al. (2004) saw an increase in
deposition velocity with increasing $NO_2$ mixing ratio; from 0.2 cm s$^{-1}$ at 1 ppbv to 0.5 cm s$^{-1}$ at 30
ppbv. Puxbaum and Gregori (1998) reported monthly averages of 0.02-0.64 cm s$^{-1}$, however, their





nighttime deposition velocities averaged below 0.05 cm s$^{-1}$. The deposition velocities determined
here are a factor of 5-40 higher than what has been measured for nighttime foliage deposition
velocities to the leaves of different trees native to California (Delaria et al., 2020; Delaria et al.,
2018), but in good agreement with measurements for daytime. It is, however, important to note
that the deposition velocities estimated here are upper limits as the estimation of the NO emission
rate is an upper limit and chemical loss of $NO_2$ is not taken into account. Using an average
nighttime $NO_2$ mixing ratio of 1650 and 1450 pptv for the Continental and Atlantic phase,
respectively, results in $NO_2$ deposition rates of 13.6 ± 6.8 μg N m$^{-2}$ h$^{-1}$ and 18.7 ± 8.5 μg N m$^{-2}$ h$^{-1}$
$^1$, which are in reasonable agreement with that measured for soil $NO_2$ deposition in a sessile oak
forest of 9.67 ± 1.92 μg N m$^{-2}$ h$^{-1}$ during the summer (Rosenkranz et al., 2006). The estimated NO
soil emission rate and $NO_2$ deposition rate are, within the uncertainties, identical, which means the
Rambouillet forest is not a significant direct source or sink of $NO_X$.

## 5   Summary:

Measurements of NO, $NO_2$, $NO_Y$, and $O_3$ during the ACROSS campaign (June-July 2022) in the
Rambouillet forest southwest of Paris, France, have been used to gain insight into nighttime
processes controlling $NO_X$ in an anthropogenically impacted forest environment. Based on
HYSPLIT back trajectories, two phases of the campaign were identified; one dominated by air
originating over the Atlantic Ocean ("Atlantic"), which on average had high relative humidity and
low $O_3$ mixing ratios, and one dominated by continental air masses from different
urban/industrialized regions ("Continental"), which on average had a lower relative humidity than
the Atlantic phase and higher $O_3$ mixing ratios. Strong diel profiles were observed in the $O_3$
measurements across the campaign with daytime peak mixing ratios varying from ~30 to 90 ppbv
and nighttime tending towards 0-10 ppbv. The daily variation was driven by a variable but
generally rapid $O_3$ deposition to soil and foliar surfaces, with a strong influence of relative
humidity (influencing the surface resistance to uptake) and inversion (influencing the rate of
entrainment of $O_3$ from above the canopy).
During the Atlantic phase, periods of sustained NO above the instrumental detection limit was
observed at nighttime when $O_3$ was sufficiently low (i.e. the NO lifetime sufficiently long). This
enabled the derivation of an average NO emission rate from the soil ($E_{NO}$) of ~1.4 ppbv h$^{-1}$, which
was confirmed by the approximately linear increase in NO observed in the absence of $O_3$ in the
Atlantic phase. The estimated $E_{NO}$ is in broad agreement with previous measurements in other
European temperate forests with tree types as found in the Rambouillet forest.
An increase in $NO_2$ at night would be expected when having a constant NO emission rate of
~1.4 ppbv h$^{-1}$ in the presence of $O_3$ as observed in this study, however, this was not the case. The
lack of increase in $NO_2$ was used to estimate first-order decay constants of $(2.0 \pm 1.0) \times 10^{-4}$ s$^{-1}$
and $(2.75 \pm 1.25) \times 10^{-4}$ s$^{-1}$ resulting in an effective lifetime of $NO_2$ of ~0.5-3 h. The loss of $NO_2$
at nighttime is presumably driven by deposition to soil and foliar surfaces since the lifetime of
$NO_2$ towards its reactions with OH and $O_3$ at night is >28 and 14 h, respectively. By comparison,
the daytime lifetime of $NO_2$ with respect to loss by reaction with OH is about 1 day. We conclude



that the nighttime deposition of $NO_2$ is a major sink of boundary layer $NO_X$ in this forested
environment.

**6    Data Availability:**
All data can be found on https://across.aeris-data.fr/catalogue/.

**7    Author contribution:**
All authors contributed with measurements. Data analysis was conducted by STA with
contributions from JNC and PD. CC and VM organized the field campaign with contributions from
the individual group leads. STA and JNC developed the manuscript with contributions from all
authors.

**8    Competing Interests:**
At least one of the (co-)authors is a member of the editorial board of Atmospheric Chemistry and
Physics

**9    Acknowledgements:**
STA is thankful to the Alexander von Humboldt foundation for funding her stay at MPIC.
PD gratefully acknowledges the Deutsche Forschungsgemeinschaft (project "MONOTONS",
project number: 522970430).
The ACROSS project has received funding from the French National Research Agency (ANR)
under the investment program integrated into France 2030, with the reference ANR-17-MPGA-
0002, and it was supported by the French National program LEFE (Les Enveloppes Fluides et
l'Environnement) of the CNRS/INSU (Centre National de la Recherche Scientifique/Institut
National des Sciences de l'Univers). Data from the ACROSS campaign are hosted by the French
national center for Atmospheric data and services AERIS.
IMT Nord Europe acknowledges financial support from the CaPPA project, which is funded by
the French National Research Agency (ANR) through the PIA (Programme d'Investissement
d'Avenir) under contract ANR-11-LABX-0005-01, the Regional Council "Hauts-de-France" and
the European Regional Development Fund (ERDF).

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



**11 Figures:**

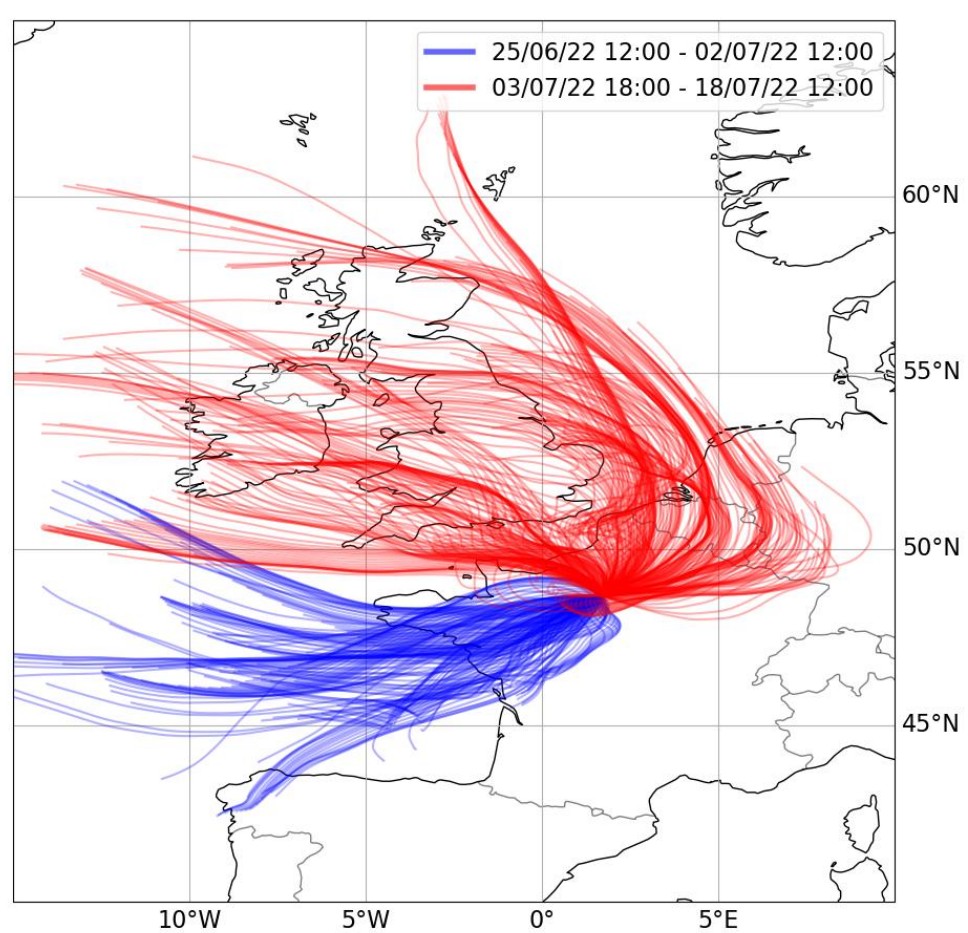

Figure 1: 48-hour back trajectories from the Rambouillet forest supersite using the Hybrid Single-Particle Lagrangian Integrated Trajectory model (HYSPLIT, version 4, 2019).

730

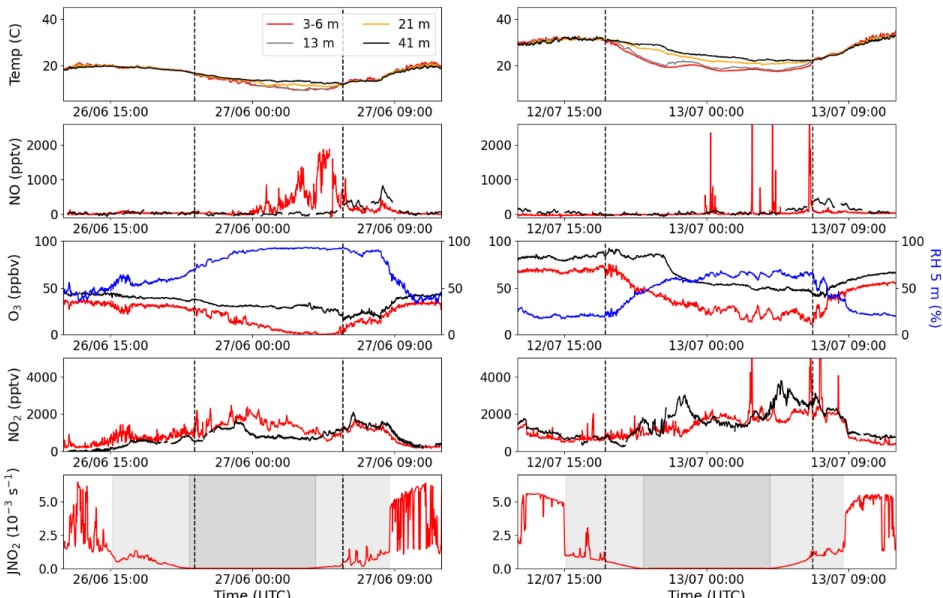

Figure 2: Measurements of temperature, NO, $O_3$, RH, $NO_2$, and $JNO_2$ for two different nights during the campaign; one during the Atlantic phase (left panels) and one during the continental phase (right panels). The different colours symbolize four different heights; red = 3-6 m, grey = 13 m, orange = 21 m, and black = 41 m, and the blue shows the RH at 5 m. The grey shaded areas in the $JNO_2$ plots shows the time the MPIC container was in shade during the afternoon and morning (light grey) and nighttime (dark grey). The vertical black dashed lines indicate the beginning and end of the observed temperature inversions in the top panels.

738



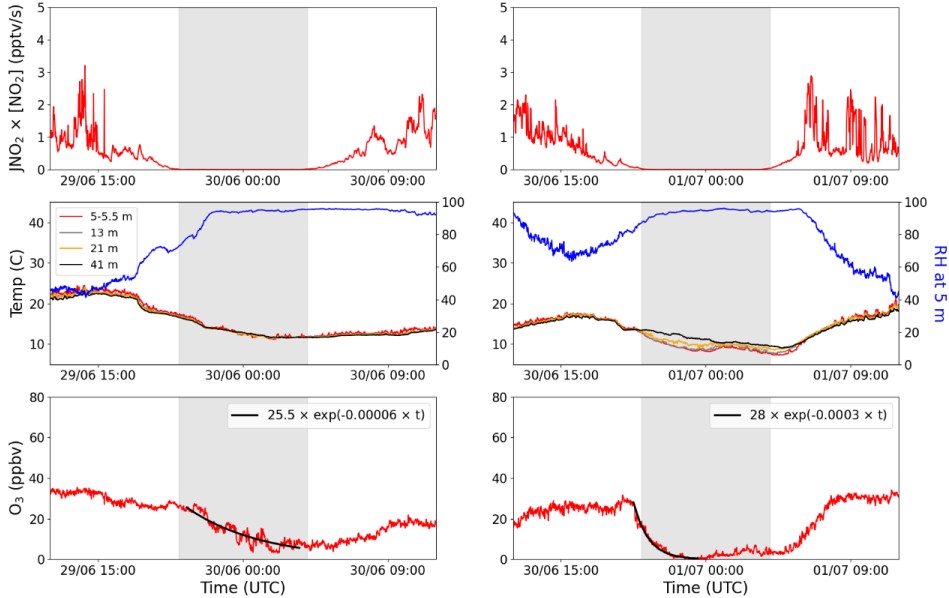

739

Figure 3: The production of O₃ (JNO₂ × [NO₂]), temperature, RH, and O₃ plotted for two nights
with high average RH; one without a temperature inversion (left panels) and one with a
temperature inversion (right panels). The different colours symbolize four different heights; red =
5-5.4 m, grey = 13 m, orange = 21 m, and black = 41 m, and the blue shows the RH at 5 m. The
net nighttime O₃ loss is fitted with an exponential decay curve (solid black line) in the bottom
plots. The grey shaded areas represent the nighttime.

746



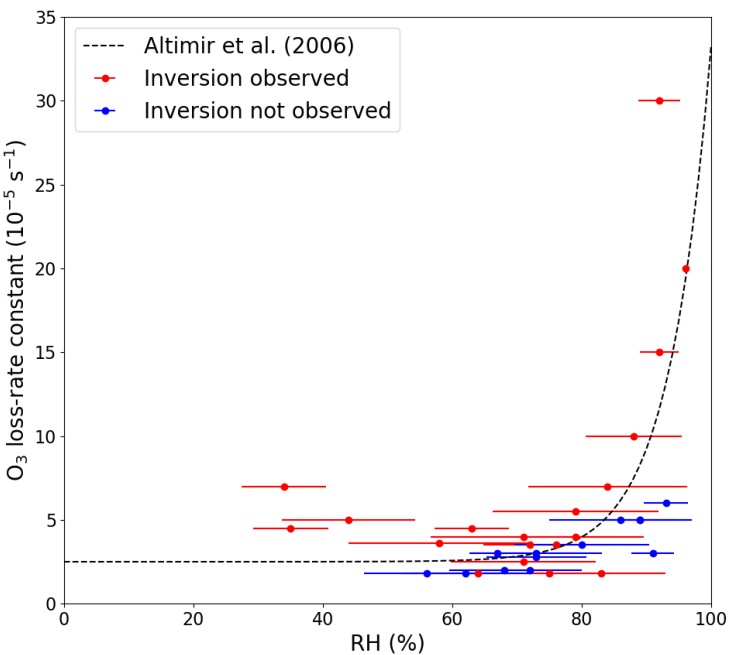

Figure 4: Net $O_3$ loss-rate constants at 5.4 m plotted against the average relative humidity measured during the time used to fit the exponential decay of $O_3$. The error bars represent ±1σ on the average RH. The dashed line symbolizes the observations made by Altimir et al. (2006).

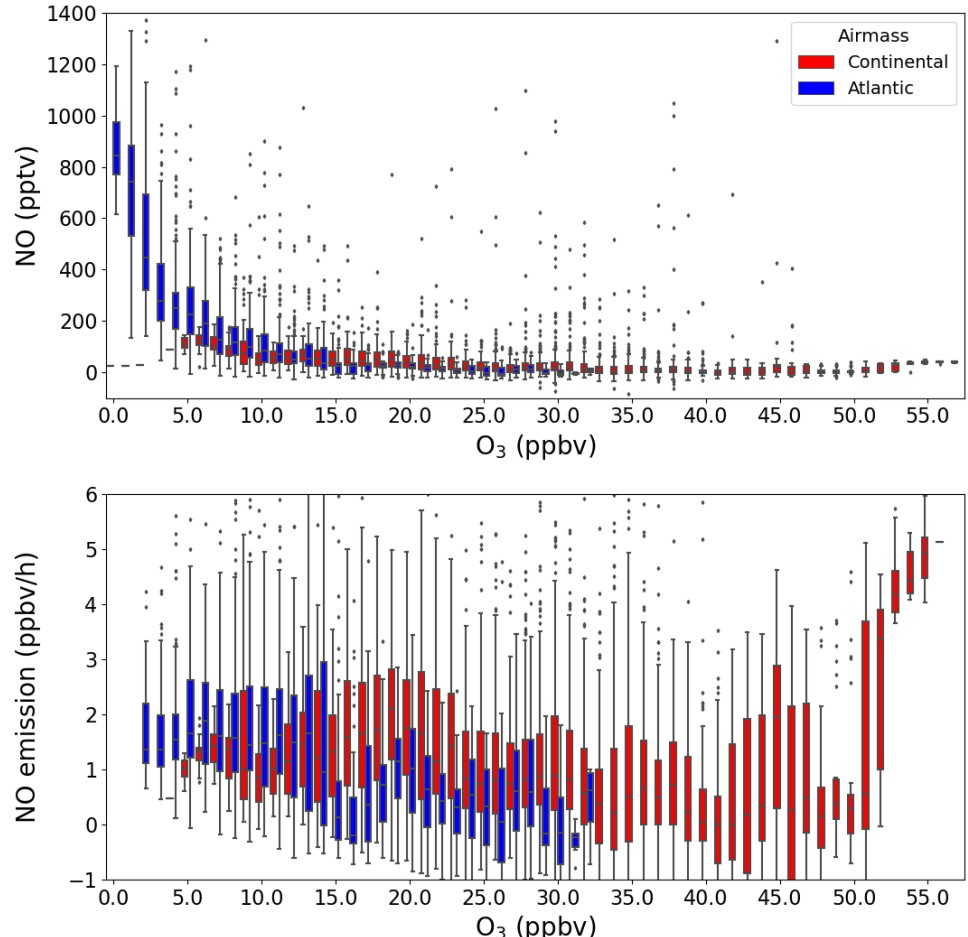

751

Figure 5: NO (top) and NO emission (bottom) plotted against $O_3$ in a box-and-whiskers plot, where
the outliers are defined as being outside $1.5 \times$ IQR. The colours represent the two different air
masses.

755





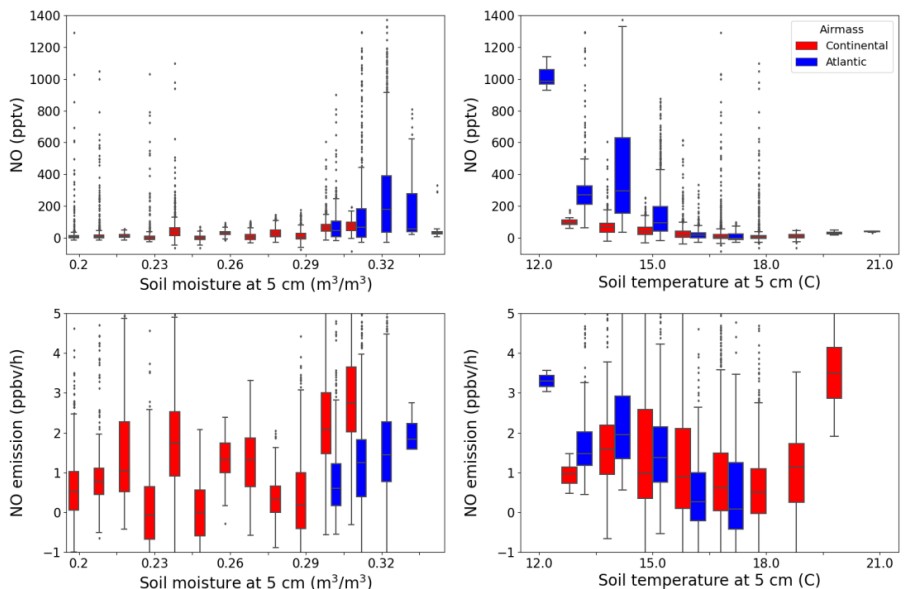

756

Figure 6: NO (top panels) and NO emission (bottom panels) plotted against soil moisture (left panels) and temperature (right panels) at 5 cm below the surface in a box-and-whiskers plot, where the outliers are defined as being outside $1.5 \times$ IQR. The colours represent the two different air masses.




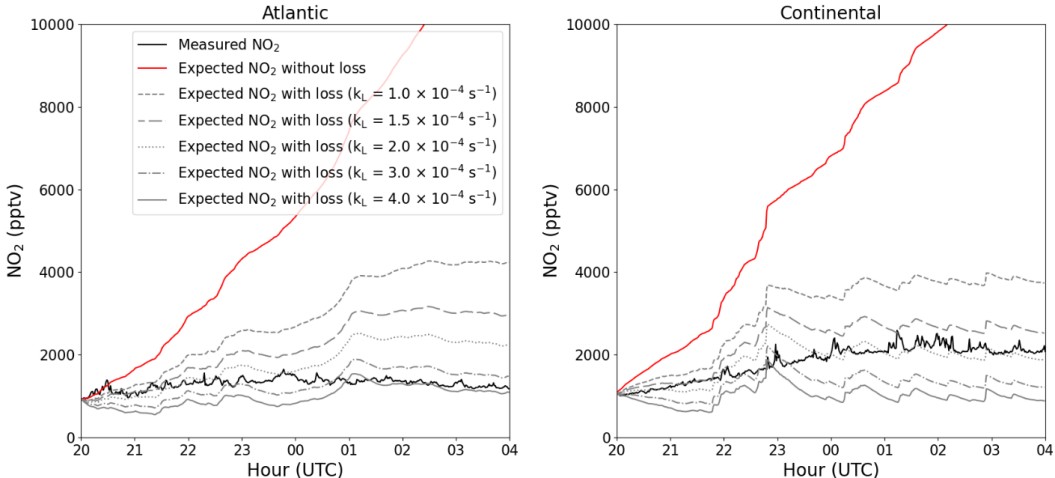


Figure 7: Average nighttime profiles of NO$_2$ at 5.4 m for each of the two phases (black) plotted
together with the expected NO$_2$ with (grey) and without (red) NO$_2$ loss.

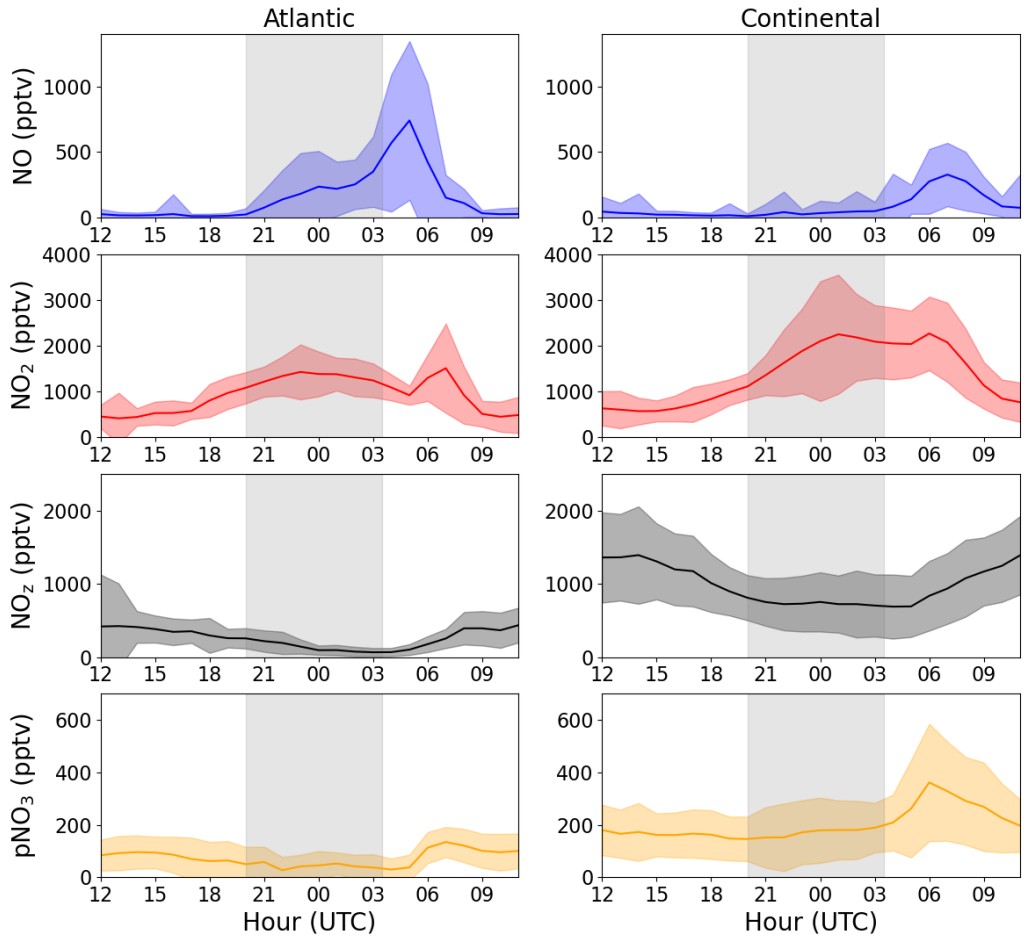

Figure 8: Average diel profiles of NO, $NO_2$, total gas-phase $NO_z$, and particulate nitrate (p$NO_3$) at 3-6 m above ground for the Atlantic (left panels) and Continental (right panels) phases. The grey shaded areas symbolize nighttime.



**12 Tables:**

Table 1: Measured NO soil emission in European forests with the same tree types as in the
Rambouillet forest.

| Dominant tree type | Location | NO emission ($\mu g$ N m$^{-2}$ h$^{-1}$) | Reference |
|---|---|---|---|
| Oak | Matra Mountains, Hungary | 2.1 <br> 6.0 ± 3.3 (summer) <br> 8.4 ± 2.4 (autumn) | (Pilegaard et al., 2006) <br> (Rosenkranz et al., 2006) <br> (Rosenkranz et al., 2006) |
| Pine | San Rossore, Italy | 5.4 | (Pilegaard et al., 2006) |
| Beech | Schottenwald, Austria | 25.5 ± 7.5 <br> 4.2 | (Schindlbacher et al., 2004) <br> (Pilegaard et al., 2006) |
| Beech | Klausen-Leopolsdorf, Austria | 10.2 ± 3.4 <br> 0.7 | (Schindlbacher et al., 2004) <br> (Pilegaard et al., 2006) |
| Spruce-Fir-Beech | Achenkirch, Austria | 2.8 ± 1.4 <br> 0.9 | (Schindlbacher et al., 2004) <br> (Pilegaard et al., 2006) |
| Mixed deciduous | Ticino Park, Italy | 18.5 ± 5.8 <br> Below LOD | (Schindlbacher et al., 2004) <br> (Pilegaard et al., 2006) |

