# Peer review of "Measurement report: Sources, sinks and lifetime of NOx in a sub urban temperate forest at night"

_EGUsphere, 2023_

## Author Comment (AC1)

**Reply to RC1 and RC2**

*In the following, the referee's comments are reproduced (black) along with our replies (blue) and changes made to the text (red).*

**General comments:**

RC1: This measurement report eloquently describes measurements of nitrogen oxides and other parameters needed to interpret factors contributing to the emission, chemistry and physical removal. The paper is exceptionally clear, the analysis clear and convincing and the interpretation persuasive. I recommend publication as is.

RC2: This study presents observations of NO, NO2, NOy, and O3 in a suburban temperate forest and highlights some important sources and sinks strongly affecting the nighttime chemistry of NOx. They found that the sustained high NO observed under very low O3 conditions could be attributed to the soil NO source, and the lack of nighttime NO2 increase can be related to its the nighttime deposition. Overall, this manuscript has comprehensively described their measurements and conducted an in-depth analysis on the observed nighttime NO2 behaviors in the Rambouillet forest environment. Although the authors didn't employ box model to support their major conclusions, I do think the current analysis looks reasonable.

We thank both reviewers for their positive assessment of our manuscript.

**Specific comments from RC2:**

I only have a few minor issues. How about aerosol concentrations during the measurements? Does the uptake of NO2 on the aerosol surface affect the NO2 removal at night?

With an NO2 uptake coefficient of less than $10^{-4}$ aerosol uptake becomes an insignificant loss term of NO2 when calculating the overall lifetime. Text has been added in line 486-488 in the revised version:

"The low aerosol surface area during ACROSS combined with the low uptake coefficient for $NO_2$ renders losses due to heterogeneous processes insignificant (IUPAC, 2024)."

I also suggest the authors to add a table to list their ground and tower measurements and this will be helpful for the readers.

We do not believe a table is necessary since it is clearly stated in the text where the different instruments were placed. All the ground measurements are described in section "3.1.1 Ground" and all measurements from the top of the tower in section "3.1.2 Tower".

Lastly, I would encourage the authors to discuss about how to reduce the uncertainties in their estimated NO emissions.

The large uncertainty we report for the calculated NO emissions is caused by atmospheric variability in e.g. NO, $O_3$, soil moisture etc. More accurate estimates of the NO emissions would require height resolved measurements of NO, $NO_2$ and $O_3$ and also direct flux NO measurements. Such measurements were not part of this particular campaign.

---

## Author Response (AR2)

Dear Editor,

Thank you for your very thorough readthrough of our manuscript.

$NO_X$, $NO_Y$, and $NO_Z$ have been corrected to $NO_x$, $NO_y$, and $NO_z$, respectively throughout the manuscript, supplementary information and figures and the header "Summary" has been changed to "Conclusions" as requested.

The abstract and conclusion have been revised according to the guidelines for ACP as described below.

Abstract:

The topic of the article, status of the scientific understanding and the gap in knowledge have been addressed in the new first sentence:

"The budget of reactive nitrogen species, which play a central role in atmospheric chemistry (e.g. in photochemical $O_3$ production), is poorly understood in forested regions."

The objectives and approach of the study are described in the second sentence:

"In this study, through observations of NO, $NO_2$, $NO_y$ and $O_3$ in the Rambouillet forest near Paris, France,

we have examined nighttime processes controlling $NO_x$ in an anthropogenically impacted forest environment."

The main results have been edited slightly from the original version to keep the abstract at 250 words and the last sentence of the abstract show the importance of the study:

"Our results indicate that the nighttime deposition of $NO_2$ is a major sink of boundary layer $NO_x$ in this temperate forest environment."

Conclusion:

In the original conclusion the results of the study were summarized, compared to previous studies and put into context by comparing the lifetime of $NO_2$ required to explain the measurements to the lifetime of $NO_2$ towards OH and $O_3$. Caveats and limitations have been added to the conclusion with the following sentences in line 535-539:

"The uncertainty in the estimated NO emission rate is determined from the uncertainties in NO and $O_3$ at 3-5 m above ground, which leads to higher relative uncertainties at low NO and $O_3$ mixing ratios. Measurements of either NO fluxes or highly resolved height profiles of NO and $O_3$ will improve the NO emission rate estimate during future field campaigns."

Best regards,

Dr. Simone Thirstrup Andersen